# Coral Resilience at Malauka'a Fringing Reef, Kāne'ohe Bay, O'ahu after 18 years

**Kelsey A. Barnhill** [1,*] and **Keisha D. Bahr** [2]

1   Faculty of Environmental Sciences and Natural Resources, Norwegian University of Life Sciences, P.O. Box 5003, 1432 Ås, Norway
2   Department of Life Sciences, Texas A&M University-Corpus Christi, Corpus Christi, TX 78412, USA
*   Correspondence: kelseybarnhill@gmail.com

**Abstract:** Globally, coral reefs are under threat from climate change and increasingly frequent bleaching events. However, corals in Kāne'ohe Bay, Hawai'i have demonstrated the ability to acclimatize and resist increasing temperatures. Benthic cover (i.e., coral, algae, other) was compared over an 18 year period (2000 vs. 2018) to estimate species composition changes. Despite a climate change induced 0.96 °C temperature increase and two major bleaching events within the 18-year period, the fringing reef saw no significant change in total coral cover (%) or relative coral species composition in the two dominant reef-building corals, *Porites compressa* and *Montipora capitata*. However, the loss of two coral species (*Pocillopora meandrina* and *Porites lobata*) and the addition of one new coral species (*Leptastrea purpurea*) between surveys indicates that while the fringing reef remains intact, a shift in species composition has occurred. While total non-coral substrate cover (%) increased from 2000 to 2018, two species of algae (*Gracilaria salicornia* and *Kappaphycus alvarezii*) present in the original survey were absent in 2018. The previously dominant algae *Dictyosphaeria* spp. significantly decreased in percent cover between surveys. The survival of the studied fringing reef indicates resilience and suggests these Hawaiian corals are capable of acclimatization to climate change and bleaching events.

**Keywords:** coral reefs; macroalgae; resilience; species composition

## 1. Introduction

Warming sea surface temperatures caused by climate change threaten coral reefs globally [1]. Increased water temperatures cause coral bleaching (reviewed in Reference [2]) which can cause total or partial mortality for colonies if the corals are unable to recover (reviewed in Reference [3]). Coral mortality leads to reef degradation as the reef loses structural complexity and is overgrown by algae, often leading to an algae-dominated phase shift [4]. Reef degradation directly causes the loss of reef-related ecosystem services, such as seafood production, shoreline protection, habitat provision, materials for medicines, and nitrogen fixation, among others [5].

Significant ecological declines driven by anthropogenic stressors are occurring on coral reefs around the world [6]. In 2000, an estimated 11% of all coral reefs had already been lost with an additional 16% damaged beyond the point of being functional ecosystems [7]. From 1985–2012 the Great Barrier Reef experienced a 50.7% decrease in coral cover [6] and the coral cover in the entire Indo-Pacific is 20% less than historical levels from 100 years ago [8]. Hawaiian reefs, however, have one of the lowest threat ratings in the Pacific (less than 30% threatened) [9]. From 1999–2012 mean Hawaiian coral cover and diversity remained stable statewide, including within Kāne'ohe Bay [10]. Reefs within Kāne'ohe Bay have repeatedly shown resilience by recovering from natural and anthropogenic disturbances such as bleaching events [11]. Increasingly frequent bleaching events threaten the longevity of coral reef

ecosystems [12] and whether or not corals can become adaptive or resistant to bleaching is contested in current literature [12]. However, corals in Kāne'ohe Bay have shown resilience through acclimatization to increased temperatures [13]. In this study resilience is defined as 'the ability of an ecosystem to recuperate its structure and functions after a perturbation' [14].

*Kāne'ohe Bay, Hawai'i*

Coral reefs in Kāne'ohe Bay, located on the northeast side of O'ahu, Hawai'i (21°4′ N and 157°8′ W), have some of the highest levels of coral cover (54–68% compared to statewide average of 24.1%) across the Hawaiian islands [10,11,15]. Due to the unique geographic properties of Kāne'ohe Bay, these reefs experience elevated summer water temperatures (1–2 °C), which offshore reefs will not be subjected to for several years [16].

Kāne'ohe Bay represents one of the few recorded examples of a phase shift reversal, in which the reefs were coral-dominant then algal-dominant and have returned to coral-dominated reefs all within a 40-year period [17]. From 1960–1970 the human population in Kāne'ohe doubled, leading to effluent municipal and military sewage to be discharged in the bay, causing eutrophication and a subsequent decline in coral cover and diversity [18]. Following the release of effluent sewage into the bay, the algae *Dictyosphaeria cavernosa*, stimulated by increased nutrient availability, spread widely, causing a phase shift from coral-dominated to algae-dominated [19,20]. Following the 1979 sewage diversion, coral cover in the bay more than doubled in just four years [21] as nutrient levels decreased [19].

The first documented coral bleaching event in Kāne'ohe occurred in 1996, in which the total coral mortality was < 1% [22]. A second, more severe bleaching event occurred in 2014 [16]. While nearly half of all corals in the southern region of the bay were pale or bleached immediately following a 2014 bleaching, there was only 1% total coral mortality three months later [23]. In 2015, another widespread bleaching event affected the Kāne'ohe Bay reefs, however a 15% decrease in bleaching compared to the 2014 event suggested some corals may be acclimatizing to increased temperatures, although higher levels of mortality were observed [11]. Kāne'ohe Bay has retained high coral cover despite Hawaiian offshore water temperatures increasing by 1.15 °C over the past 60 years [11]. Corals within the bay also show increased resistance to acidification and warming waters compared to other corals in O'ahu [24]. The historical resilience of corals in Kāne'ohe Bay and the consistently high coral cover while many reefs around the globe are in decline led to the following research question: How has coral cover and community composition changed in response to 18 years of warming temperatures and two major bleaching events in a well-studied coral reef ecosystem?

## 2. Materials and Methods

### 2.1. Study Site: Kāne'ohe Bay, Hawai'i

The study site was a 600-meter section of the Malauka'a fringing reef (21.44300899° N, 157.80636° W to 21.43853104° N, 157.806541° W) in the south-west of Kāne'ohe Bay, which was initially surveyed in 2000 [25]. Similar to other reefs in the bay, *Porites compressa* and *Montipora capitata* are the dominant reef-building corals. The northern section of the reef is approximately 125 meters offshore of Kealohi Point at He'eia State park. The southern 200 meters of the study site is adjacent to the Paepae o He'eia (traditional Hawaiian fishpond) where there is ongoing estuarine restoration focusing on sociocultural benefits [26]. The southern end of the reef is subjected to freshwater stream and pond output from He'eia stream and a triple mākāhā (sluice gate) within Paepae o He'eia [27]. The selected reef suffered bleaching and low mortality (<5%) during the 2014/2015 bleaching event [11].

## *2.2. Comparative Study Setup*

### 2.2.1. Benthic Survey

Coral cover and benthic community composition were measured through a quali-quantitative comparison using a modified version of the point intercept transect (PIT) (as described by Reference [28]) in the initial survey (2000) and follow up survey (2018). The PIT method identifies benthic cover every 50 cm along a transect [29]. During the 2000 study [25], benthic cover was recorded every meter and thus repeated as such in the 2018 study. Coral species, algae species, crustose coralline algae, turf, sand, and rubble were recorded along each transect. Crustose coralline algae and turf were pooled together into 'non-coral substrate' and sand and rubble were pooled together into 'mixed sand' as they were not separated from one another in the 2000 survey. Additionally, transects from the 2000 study continued until the edge of the reef platform was reached, causing transects to consist of varying lengths (5–34 m) dependent on the width of the reef. The locations of transect sites (n = 60) during the 2000 survey were resurveyed in 2018 using a Garmin GPSMAP 78s; 3 m accuracy (Garmin Ltd., Olathe, KS, USA). Transects were spaced 10 meters apart to survey the 600-meter portion of the fringing reef (Figure 1). Both surveys were conducted with one snorkeling observer identifying all species in situ. Two community descriptors, cover and community composition, are used to empirically describe resilience to environmental stressors present at the site [14].

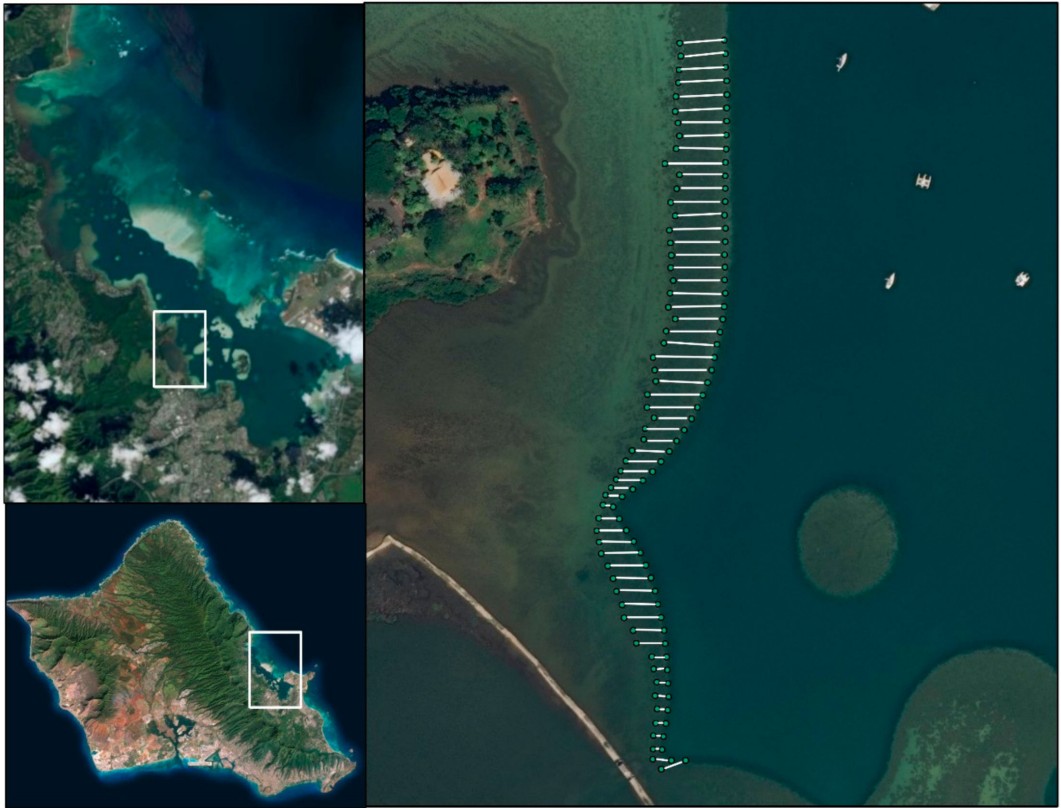

**Figure 1.** Map of Malauka'a fringing reef with transects overlaid within Kāne'ohe Bay, O'ahu. Note the variation in transect length due to reef width. Photo Credit: Digital Globe.

### 2.2.2. Seawater Temperature

Daily mean seawater temperatures (°C) for 2000 to 2018 in Kāne'ohe Bay were calculated from PacIOOS Moku o Lo'e weather station (http://www.pacioos.hawaii.edu/weather/obs-mokuoloe/).

2.2.3. Statistical Analysis

A two-tailed t-test was used to determine changes in daily average temperatures between 2000 and 2018 within RStudio IDE Version 1.1.456 (RStudio, Inc., Boston, MA, USA) [30]. A non-metric multidimensional scaling (NMDS) ordination plot using Bray–Curtis distance was created to visualize the 2000 and 2018 benthic communities within ggplot in RStudio [30]. A matched pair Wilcoxon signed-rank analysis was used to compare changes in individual species and groups (i.e., corals, algae, and mixed sand) between years (2000 vs. 2018) within transects using JMP Pro 13 (*JMP®*, Version 13, SAS Institute Inc., Cary, NC, USA) [31]. A permutational multivariate ANOVA (PERMANOVA) and a permutational test of multivariate dispersion (PERMDISP) were ran to determine if overall species composition changed between 2000 and 2018 using PERMANOVA+ [32] in PRIMER 7 Version 7.0.13 (PRIMER-e (Quest Research Limited) Auckland, New Zealand) [33]. The data for the PERMANOVA and PERMDISP was square root transformed before calculating a Bray–Curtis similarity matrix. The PERMANOVA was ran with two factors- fixed factor 'year' (2 levels, 999 unique permutations) and random 'transect' (6 levels, with transects pooled into 6 groups of 10 based on location, 998 unique permutations) nested in 'year.'

## 3. Results

### 3.1. Benthic Survey

Transects ranged from 5 to 34 meters in length, with 1219 observations recorded at one-meter intervals along the fringing reef in both 2000 and 2018. Six species of coral (i.e., *Porites compressa, Porites lobata, Montipora capitata, Lobactis* (formally *Fungia*) *scutaria, Pocillopora damicornis, Pocillopora meandrina*) were recorded at the site in 2000 and four (i.e., *P. compressa, M. capitata, P. damicornis, Leptastrea purpurea*) were recorded in 2018. Four species of macroalgae (i.e., *Dictyosphaeria cavernosa, Dictyosphaeria versluyii, Gracilaria salicornia, Kappaphycus alvarezii*) were present in 2000 and two (i.e., *D. cavernosa, D. versluyii*) were present in 2018. Unidentified species of turf algae, crustose coralline algae, and mixed sand and rubble were present in both surveys and were marked as such.

### 3.2. Statistical Analysis

3.2.1. Abiotic and Biotic Changes

The mean daily temperature (mean ± SE) at Moku o Lo'e increased from 24.07 ± 0.07 °C in 2000 to 25.03 ± 0.02 °C in 2018 ($p < 0.0001$), despite no evident general trend across years ($R^2$=0.1852) (Figure 2). The overall community composition across the fringing reef changed from 2000 to 2018 (PERMANOVA $p < 0.05$, PERMDISP $p < 0.05$) (Figure 3, Table 1). Total mixed sand cover decreased significantly from 12 ± 1.9% to 4.6 ± 1.0% from 2000 to 2018 ($p < 0.0001$) (Figures 4 and 5). This is further supported by a break in the fringing reef in 2000 (represented as a transect with 100% sand cover), which was not observed in the 2018 survey.

**Table 1.** PERMANOVA model results based on a Bray–Curtis similarity matrix comparing benthic communities between years (fixed factor) and transect section (random factor nested within year). Significant p values ($p < 0.05$) are bolded.

| PERMANOVA | | | | | | | PERMDISP | |
|---|---|---|---|---|---|---|---|---|
| **Source** | **df** | **SS** | **MS** | **Pseudo-F** | **_p_-Value** | **Unique Perms** | **F** | **_p_-Value** |
| Year | 1 | 25896 | 25896 | 5.3004 | **0.003** | 999 | 11.806 | **0.004** |
| Transect (Year) | 10 | 48975 | 4897.5 | 9.5632 | **0.001** | 998 | 9.8724 | **0.001** |
| Residuals | 108 | 55308 | 512.11 | | | | | |
| Total | 119 | 1.3004E+05 | | | | | | |

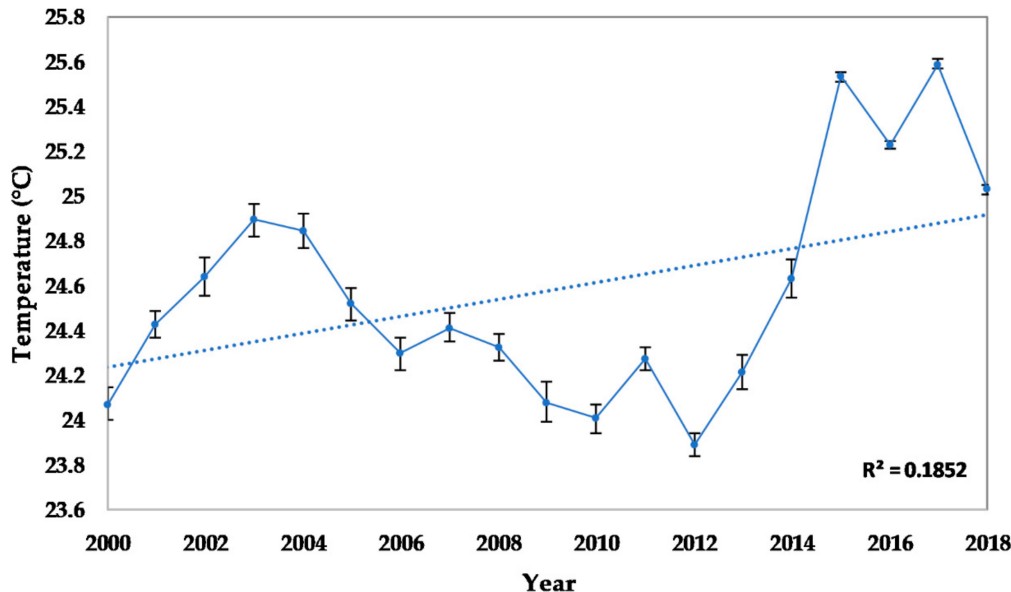

**Figure 2.** Average annual daily mean seawater temperature (°C) at Moku o Lo'e (Coconut Island) at the Hawai'i Institute of Marine Biology from 2000–2018. Data retrieved from (http://www.pacioos. hawaii.edu/weather/obs-mokuoloe/).

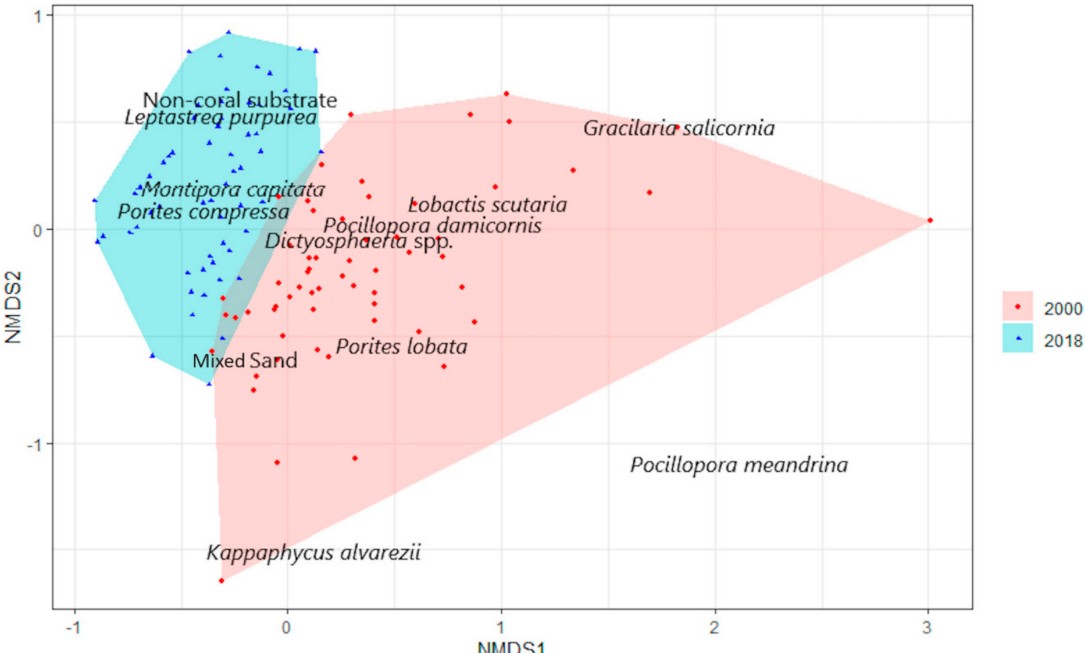

**Figure 3.** Non-metric multidimensional scaling (NMDS) ordination plot representing the benthic communities from the 2000 and 2018 surveys in convex hulls (Dimensions = 2, Stress = 0.19). Each point represents one transect (n = 60).

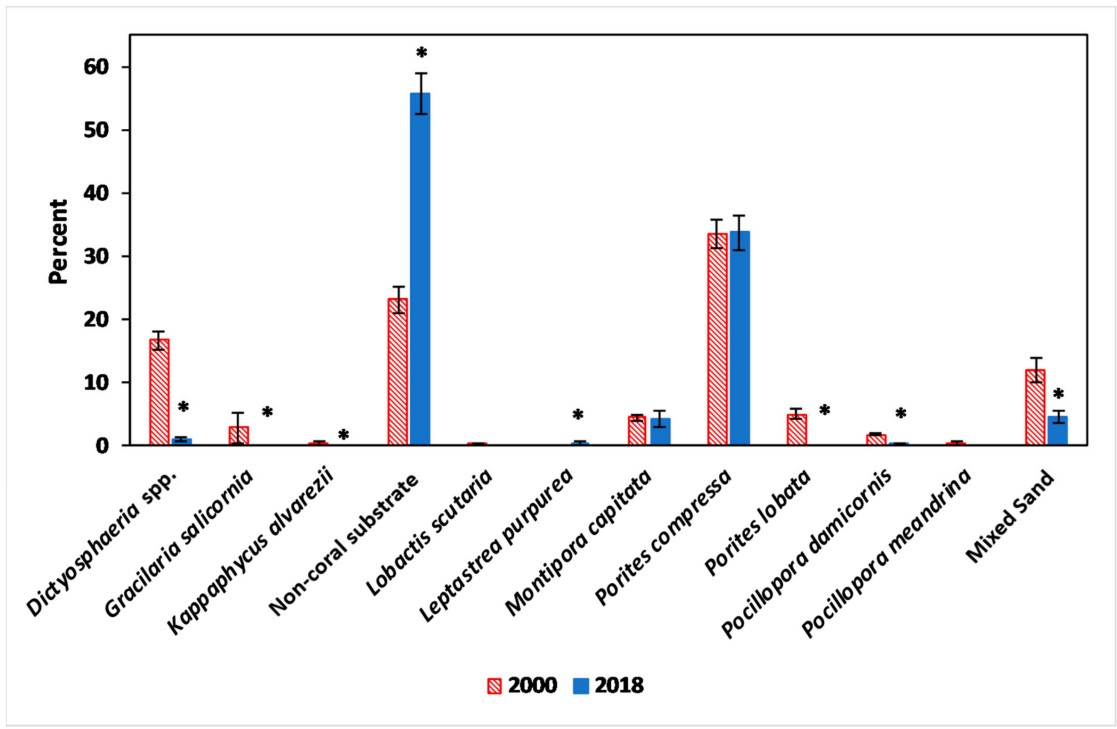

**Figure 4.** Mean Percent cover of each species or category in 2000 vs. 2018. Each standard error bar is one standard error from the mean. * Indicates significant difference between years at $p < 0.05$.

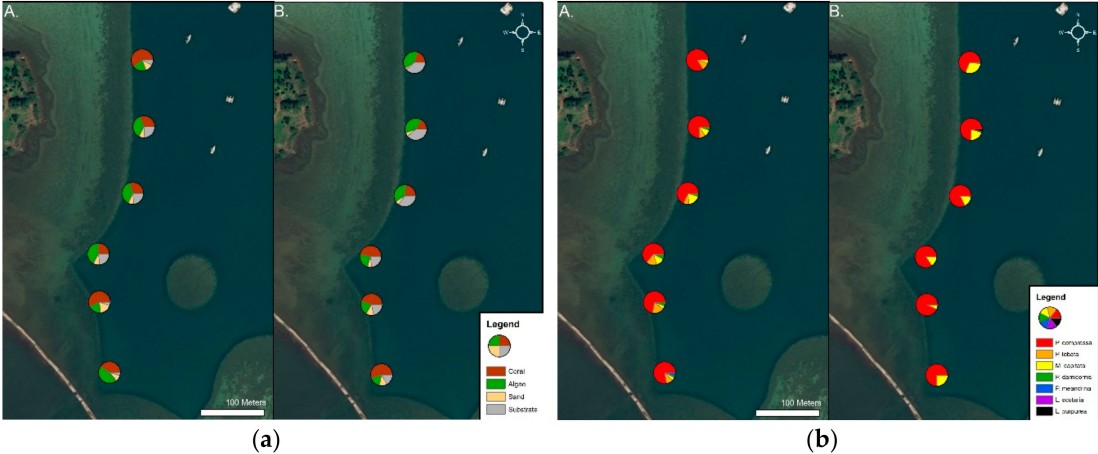

<div align="center">(a)         (b)</div>

**Figure 5.** Spatial trends in (**a**) Total benthic cover and (**b**) coral species composition. (**A.**) represents data from the 2000 survey and (**B.**) represents data from the 2018 survey. Each pie chart represents the average from 10 transects in that section. Photo credit: Digital Globe

### 3.2.2. Algae

The total overall algae cover across the entire site increased significantly from $42.9 \pm 3.1\%$ in 2000 to $56.8 \pm 3.2\%$ in 2018 ($p = 0.0009$) (Figures 4 and 5). *Dictyosphaeria* spp. (*D. cavernosa* and *D. versluyii*) decreased significantly from $16.7 \pm 1.5\%$ in 2000 to $1.1 \pm 0.3\%$ 2018 ($p < 0.0001$). *Gracilaria salicornia* and *Kappaphycus alvarezii* were both present in 2000 ($2.8 \pm 2.4\%$, and $0.33 \pm 0.3\%$ respectively) and absent from the 2018 survey ($p = 0.0002, 0.045$). Non-coral substrate (turf, crustose coralline algae) increased significantly from $23.1 \pm 2.1\%$ in 2000 to $55.6 \pm 3.2\%$ in 2018 ($p < 0.0001$).

### 3.2.3. Coral

Total coral cover did not change between 2000 (45.1 ± 2.5%) and 2018 (38.6 ± 2.9%) (matched pair Wilcoxon signed-rank; $p$ = 0.0810) (Figures 4 and 5). Neither dominant reef-building species (i.e., *Porites compressa* nor *Montipora capitata*) experienced a significant change in coverage percent. *Porites compressa* was found to cover 33.6 ± 2.3% and 33.7 ± 2.8% of the reef ($p$ = 0.8784) and *M. capitata* was found to cover 4.4 ± 0.6% and 4.2 ± 1.2% ($p$ = 0.7836) in 2000 and 2018, respectively. *Porites lobata* (5 ± 0.8%, $p$ < 0.0001), *Pocillopora meandrina* (0.16 ± 0.4%, $p$ = 0.1590), and *Lobactis scutaria* (0.16 ± 0.1%, $p$ = 0.1590) were all present in the 2000 survey, but absent in 2018. *Lobactis scuatria* was visually observed at the site; however, it was not present on survey transects (personal observation, K.A.B., July 2018). *Pocillopora damicornis* decreased significantly from 1.8 ± 0.3% to 0.25 ± 0.1% from 2000 to 2018 ($p$ = 0.0005). *Leptastrea purpurea* was not present in the 2000 survey but represented 0.49 ± 0.3% of the total cover in 2018 ($p$ = 0.0241).

Spatial variations between 2000 and 2018 were also observed (Figure 5). The percent of coral cover was consistent between sections of the reef in 2000, whereas the percent of coral cover increased at the southern portion of the reef in 2018. In 2018, non-coral substrate was most common at the northern section of the reef, whereas it was more evenly distributed in 2000. *Montipora capitata* prevalence also increased in the southern portion of the reef from 2000 to 2018.

## 4. Discussion

While many reefs globally are in decline due to anthropogenic factors, coral cover on the reefs in Hawai'i remained stable from 1999–2012 [10]. Returning to the Malauka'a fringing reef provided an opportunity to explore decadal change in coral cover across an entire 600-meter reef. Results of this study revealed resilience and stability at the Malauka'a fringing reef over the past 18 years compared to other reefs across the Hawaiian islands. We predict the reef will show the same resilience as most reefs in Kāne'ohe Bay through maintaining high coral cover in the face of climate change.

### 4.1. Abiotic and Biotic Changes

During the 18 years between the two survey periods, corals at the study site experienced two consective bleaching events (i.e., 2014 and 2015). Seawater temperatures during these periods exceeded 31 °C for several days with cumulative heating of five degree heating weeks (DHW) in 2014 and 12 DHW in 2015 [11]. Between 2000 and 2018, daily average temperatures increased by 0.96 °C in Kāne'ohe Bay, indicating higher levels of temperature stress in 2018 compared to 2000.

The significant decrease in percent cover of mixed sand indicates the proportion of live benthic cover expanded between surveys.

### 4.2. Algae

*Dictyosphaeria cavernosa* was once the dominant algae species in Kāne'ohe Bay, responsible for one of the first well-studied reef phase shifts from coral-dominated to algae dominated [20]. Following the phase-shift reversal, the algae persisted in the bay due to overfishing of herbivorous fish that would have placed grazing pressure on the species [20]. *Dictyosphaeria cavernosa* remained abundant in Kāne'ohe Bay, averaging 16% total cover during a 1996–1997 survey [20]. The findings of the 2000 survey indicate the percent cover of *Dictyosphaeria* spp. remained at a comparable level three years later at the fringing reef (16.7 ± 1.5%). In 2006, following an unusually rainy period, decreased irradiance combined with slow spring growth rates for the species caused *D. cavernosa* to effectively disappear from Kāne'ohe Bay [34]. Immediately following the rapid decline, reefs nearby Moku o Lo'e averaged 0–4% total cover of *D. cavernosa* [35]. In 2018, twelve years later, the prevalence of *D. cavernosa* has remained greatly diminished at this fringing reef (1.1 ± 0.3%), suggesting an enduring phase shift reversal.

The invasive species *G. salicornia* was introduced to Kāne'ohe Bay in the 1970's and quickly spread, overgrowing and smothering reef-building corals [36]. The invasive algae has since decreased over the past few years as a result of biocontrol [37], manual removal [38], and increased grazing from *Chelonia mydas,* the green sea turtle [39]. The management efforts and return of *C. mydas* to Kāne'ohe Bay likely explain why the once dominant macroalgae was not observed during the 2018 survey.

Like *G. salicornia*, *Kappaphycus alvarezii* (formerly *Eucheuma striatum*) was introduced to Kāne'ohe Bay in the 1970's [40] and had spread across the southern and central bay by 1996 in a near-cosmopolitan distribution [41]. A total percent cover of 0.33 ± 0.3% in the 2000 survey was slightly higher than the mean 0.06 ± 0.02% cover found at four shallow fringing reefs in the central bay in 1996 [41]. Amidst fears of further spreading, preliminary management options for *Kappaphycus* spp. were assessed in 2002 [42]. Divers used an underwater vacuum cleaner and outplanted juvenile urchins (*Tripneustes gratilla*) to remove and control the species in 2011–2013, leading to an 85% decrease in invasive macroalgae across sites [38]. Management efforts have continued to be successful as *K. alvarezii* was not observed at the study site during the 2018 survey.

Despite *Dictyosphaeria* spp., *G. salicornia,* and *K. alvarezii* all decreasing or disappearing from the reef, a total increase in algal cover from 2000 to 2018 was observed, mainly due to the increase in 'non-coral substrate'. It should be noted that 18.6 ± 0.8% of the non-coral substrate from the 2018 survey was crustose coralline algae (CCA). CCA was not categorized or differentiated from 'encrusted corals' in the 2000 study. Thus, the percent cover of total algae as well as non-coral substrate is inflated in the 2018 data and likely the 2000 data as well. Unlike turf and macroalgae, CCA promotes coral recruitment and recovery [43] and would have ideally been separated into its own category.

The high percentage of non-coral substrate in 2018 (55.6 ± 0.9%) was also impacted by the prevalence of (perhaps short-lived) turf on the tips of *P. compressa* and *M. capitata.* The tips of these reef-building corals were susceptible to warming events and air exposure at extreme low tides as the 2018 survey was conducted in late July following a warm period and spring tides (Figure 6). Observed spatial differences within benthic communities showed certain sections of the reef were more susceptible to algal growth. During the 2018 survey, the northern portion of the reef exhibited higher levels of non-coral substrate than the southern portion of the reef (Figure 5). In addition to spatial variations in low tide air exposure, differences in temperatures could explain this occurrence as corals near the northern end experienced increased thermal stress (2018 summer midday average (11:00–16:00) temperature 27.72 ± 0.94) compared to corals at the southern end (2018 summer midday average temperature 27.48 ± 0.96). This difference highlights the importance local microclimates have on coral communities.

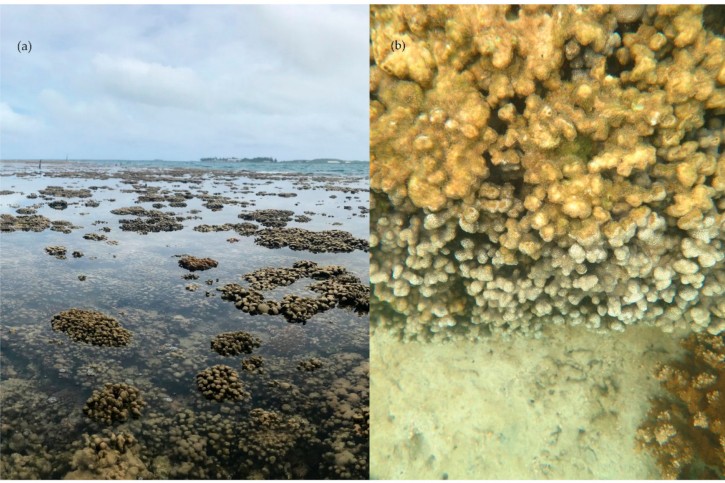

**Figure 6.** Reef Air Exposure. (**a**) Reef exposed during low tide in Kāne'ohe Bay (Picture Credit: KDB). (**b**) Tips of a pale *P. compressa* colony covered with turf (Picture Credit: KAB).

### 4.3. Corals

Despite a significant increase in algal cover between surveys, total coral cover was similar in 2000 and 2018. *Porites compressa* sustained a high percent cover over 18 years at the fringing reef despite decreasing in percent cover by 22.9% in 14 years (1999–2012) across the Hawaiian Islands, with significant declines on the island of Oʻahu [10]. *Porites compressa* is known to be sensitive to increased temperatures, which can cause bleaching and decreased calcification rates for the species [44]. Despite temperature increases over the 18 years, *P. compressa* has maintained its dominance as the most prevalent coral species at Malaukaʻa fringing reef, supporting its ability to acclimatize and persist in warming waters [24].

The *Montipora capitata* percent cover remained at a similar level between surveys despite increasing in percent cover by 56.8% in 14 years (1999–2012) across the Hawaiian Islands [10]. However, this study extended transects only to the end of the continuous reef pavement and many *M. capitata* colonies were located inshore of the reef (personal observation, K.A.B., July 2018). *Montipora capitata* colonies in Kāneʻohe Bay have shown resilience through the ability to acclimatize/adapt to temperature increases (2.6 °C) over the past 47 years [13]. The continued presence of *M. capitata* at Malaukaʻa fringing reef despite temperature increases supports the findings of [13] through indicating resilience in lab experimentation and field long-term monitoring.

Percent cover of *Pocillopora damicornis* decreased significantly between the 18 years. The species is known to be highly sensitive to decreased salinity levels [15]. Increased freshwater input onto the southern portion of the surveyed reef may have impacted the abundance of *P. damicornis*. Following biocultural restoration of the Paepae o Heʻeia, water exchange between the fishpond and the adjacent reef increased, with an additional 14,418 m$^3$ of pond water being flushed out onto the reef during each ebb tidal cycle [27].

In 2000, *P. lobata* was a common reef-building coral at the study site. However, *P. lobata* was not observed in the 2018 survey. *Porites lobata* was described as 'common to Kāneʻohe Bay' in 1999 [45]; however, more recently it was estimated to have 0–1% cover along Kāneʻohe's fringing reefs [46,47]. Previous work suggests that *P. lobata* and *P. compressa* are different morphotypes of the same species and/or hybridize frequently [48]. Therefore, the disappearance of *P. lobata* may mean one morphospecies was selected over the other. Due to similarities between *P. lobata* and *P. compressa* as well as the possibility of hybridizations, there may be potential misidentifications in the 2000 survey.

Similar to *P. lobata*, *P. meandrina* was also estimated to have 0–1% cover along fringing reefs in Kāneʻohe Bay, supporting its absence in the 2018 survey [46,47]. *Pocillopora meandrina* has been similarly decreasing in percent cover across the Hawaiian Islands, with a 36.1% decrease from 1999–2012 [10]. Following the 2015 bleaching event, 98% of *P. meandrina* colonies on the west side of the island of Hawaiʻi were partially or fully bleached, demonstrating they are one of the more susceptible species to thermal stress [49]. They were similarly listed as the least resistant species to thermal stress at Kahe Point, Oahu [50]. The species vulnerability to increased temperatures may explain its disappearance in the 2018 survey.

*Lobactis* (formely *Fungia*) *scutaria* was recorded during the 2000 survey but not observed in the 2018 survey. Low densities of *L. scutaria* are expected at the site, as the species is abundantly found on patch reefs in Kāneʻohe Bay, not fringing reefs [51]. Future studies of the area should employ a survey method such as the 'quadrat method', which avoids sampling from a small number of points to ensure rare and very rare species are included [28].

*Leptastrea purpurea* was the only new species seen in the 2018 survey. This encrusting species is tolerant to elevated temperatures and has been seen in areas where other coral species have succumbed to thermal stress [50]. The hardy species has been declared one of the 'long-term winners' as *L. purpurea* increase in abundance during thermal stress events [52,53]. *Leptastrea purpurea* has a relatively low metabolic rate, a characteristic known to help corals tolerate high temperatures [54]. Increasing temperatures may have allowed *L. purpurea* to settle in an area it had not before been present in, as it now holds a competitive advantage over other species which are less tolerant to thermal stress [53].

Coral cover did not significantly change over the past 18 years, although temperatures increased by 0.8 °C and two bleaching events (2014 and 2015) occurred during that time frame. While the fringing reef has shown resilience, it is unclear whether or not acclimatization and resistance to climate change has impacted its success. Previous work [13] has found all three species (i.e., *M. capitata*, *L. scutaria*, *P. damicornis*) of Hawaiian corals tested within Kāne'ohe Bay have higher survivorship at 31 °C today than they did in 1970, suggesting that these corals can adapt to higher temperatures. As the corals in this study were from similar locations as those used by References [13] and [24], it is possible the resilience seen on the reef can be attributed in part to adaptation or acclimatization. The persistence of the coral cover at this site occurred while other sites within Kāne'ohe Bay decreased in coral cover. From 2012–2016, Hawaii Coral Reef Assessment & Monitoring Program (CRAMP) reef sites at He'eia and Moku o Lo'e decreased by 19.7% and 42.2%, respectively [11].

However, while the total coral cover remained relatively stable over the past 18 years, the species composition has changed. The decrease in the total number of coral species present in the survey (6 in 2000, 4 in 2018) represents an overall loss in biodiversity. Additionally, two (or one if *P. lobata* is considered to be the same species as *P. compressa*) species of coral were lost in the 18 years while one non-reef building coral (*L. purpurea*) was added. This change suggests a temperature-driven shift in species composition over the 18 years. While the total coral cover remains high, the loss of locally uncommon species has negative impacts as rarer species often support more vulnerable and unique ecosystem functions [55].

Despite a shift in coral species composition, total coral cover percent remained unchanged over the 18 years and populations of the two dominant species of coral remained at comparable levels. Despite evidence of Hawaiian coral adaptation to increased temperatures, this adaptation might not occur fast enough to tolerate projected increasingly frequent bleaching events [13]. While the Malauka'a fringing reef has shown resilience over the past 18 years, the amount of warming and the rate of temperature increase will determine the fate of these reefs.

**Author Contributions:** Conceptualization, K.A.B. and K.D.B.; methodology, K.A.B. and K.D.B.; software, K.A.B. and K.D.B.; validation, K.A.B. and K.D.B.; formal analysis, K.A.B.; investigation, K.A.B.; resources, K.D.B.; data curation, K.A.B. and K.D.B.; writing—original draft preparation, K.A.B.; writing—review and editing, K.D.B.; visualization, K.A.B. and K.D.B.; supervision, K.D.B.

**Funding:** K.A.B acknowledges NMBU's Noragric field stipend for travel support to conduct research and NMBU's Publishing Fund for supporting the article processing charges. K.D.B. also acknowledges NSF#OA14-16889 for salary support during the writing and publication process.

**Acknowledgments:** The authors would like to thank Ku'ulei Rodgers for field advice and Ian Bryceson for his feedback on early drafts. Thanks to Annette Breckwoldt for providing data from the original 2000 survey. Thanks to Snorre Sundsbø and Elildo AR Carvalho Jr for assistance in statistical analyses. Thanks to Ashley McGowan, Colleen Brown, Robert Barnhill and members of the Coral Reef Ecology Lab for field and logistical support. Water temperature data was provided in part by PacIOOS (http://www.pacioos.org), which is a part of the U.S. Integrated Ocean Observing System (IOOS®), funded in part by National Oceanic and Atmospheric Administration (NOAA) Award ##NA16NOS0120024. We also thank three anonymous reviewers. We appreciate the time that each has invested to provide the review, and these comments have helped us improve the manuscript significantly.

**Conflicts of Interest:** The authors declare no conflict of interest.

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
