# Peer review of "Coral Resilience at Malauka`a Fringing Reef, Kāneʻohe Bay, Oʻahu after 18 years"

_jmse, doi:10.3390/jmse7090311_

Round 1

Reviewer 1 Report

The ms by Barnhill and Bahr reports on the results of a re-visitation of a fringing reef in Hawai’i after 18 years. It provides novel and useful information and I recommend publication, provided that a number of changes are introduced. Both in the title and through the text, the ms refers to the idea of resilience. Apparently, the authors measured resilience in terms of quali-quantitative composition of the reef benthic community. This should be clearly stated. Quoting a general review about coral reef resilience would also be useful (see, for instance, Bianchi C.N. et al., 2017. Resilience of the marine animal forest: lessons from Maldivian coral reefs after the mass mortality of 1998. In: Rossi S., Bramanti L., Gori A., Orejas C., eds. Marine Animal Forests. Springer, Cham, DOI https://doi.org/10.1007/978-3-319-21012-4_35). In addition, evaluating resilience using re-visitation data only may be questionable. The authors should discuss while they did not consider the idea of resistance: if there was no strong reduction of coral cover in the years in between, no recovery would had been necessary.

Minor remarks

Line 21 – spp. should be written in Roman, not Italics.

Lines 47-73 – the whole section 1.1. Kāneʻohe Bay, Hawaiʻi seems misplaced in the Introduction. I suggest moving to Materials and Methods and merged with 2.1. Study Site.

Line 85 – I wonder about the use of the world ‘Experimental’. This is not an experimental study, but rather a descriptive/comparative study.

Lines 130-132 – How the ellipses of Figure 2 have been calculated is not clear: are they centred on year centroids? Why their axes are not parallel to MDS axes? A short explanation is strongly needed: for instance, it can be added within line 108 after [29].

Figure 3 - spp. and Mixed Sand should be written in Roman, not Italics.

Figure 4 – Legends are unreadable.

Line 161 – Here the term resiliency is used, while in the title and through the text resilience is used. I suggest using always the same term.

Line 189 – I understood that there is only ‘green sea turtle’, so ‘a green’ should be replaced with ‘the green’.

Line 196 - spp. should be written in Roman, not Italics.

Line 197 – I would recommend adding ‘cleaner’ after ‘vacuum’ to be clearer.

Line 201 - spp. should be written in Roman, not Italics.

Line 229 – ‘personal observation’ by whom? both authors? and when exactly?

Line 245 – I suggest deleting the phrase ‘and there has not been a decrease in biodiversity for Porites’ as it is vague and unbacked by any evidence. Based on what you say, there are two possibilities: i) one morphotype has been completely replaced by the other; ii) the two morphotypes were present in 2000 and only one remains in 2018 (this might imply a loss of genetic biodiversity).

Line 287 – ‘biodiversity’ is not the best term here; maybe ‘composition’ might be better.

References – The use of capital letters in the titles is not consistent through the list. Latin names of the species should always be written in Italics.

Reviewer 2 Report

After reading this manuscript several times, I am left with an overall concern with the conclusions of the study which ties into the temporal nature of the survey methodology. While the two independent sampling events utilize a seemingly standard (despite the 1m versus 50cm sampling interval on the transects per the methodology stated) approach for both events (2000 and 2018) it is not valid to assume a trend as suggested in the abstract and conclusion. Two data points/sets do not make a trend. They are certainly important data sets from this location but it would be a much stronger paper if they had interim sampling - particularly before and after the two "major" bleaching events between the 2000 and 2018 sampling periods. The species composition shifts (particularly algae) may simply be due to larger scale gradients or phenomena in the estuary complex. Discussion and consideration of the larger system dynamics would have also been important to include to couch the results from the study site.

There is just to much of a leap here and conclusion inference between the two sampling periods. I see this as a major flaw that can only be remedied by additional sampling on a more regular basis

That being said, coral is obviously a long lived species so the more important story here is the species shift and persistence despite the overall increase in water temperature across the time period. That fact may not warrant a full paper submission but certainly a note in a future meta-analysis potentially.

The graphic representation was of very high quality on a positive note.

Reviewer 3 Report

The paper reports the comparison of macroassemblages on a coral reef between the two replicate surveys 2000 and 2018. The design of field surveys seems appropriate, and the results are interesting, showing no decrease in dominant corals' cover despite the apparent general trend  induced by climate change. The statistical analysis of data, however, is inappropriate and should be corrected prior to publication (which may potentiall alter some results and conclusions). The Discussion could be shortened and less speculative. See the detailed comments below (the most important are on PERMANOVA design, pairwise species-specific comparisons, and Figures 2 and 4):

Title: "Decadal trends" should be toned down, since there are only two points in time.

Abstract: Looks like the main result is that what is called "non-coral substrate" overtook two species of algae and mixed sand (see Fig. 3), but this is not covered in abstract; significance

of minor changes to coral covers is likely a false-positive resulting from inappropriate statistical procedure (see below).

l87-90 I expect the detailed explanation of the sampling procedure instead of references. For  instance, from the Results I guess that not only the covers of live species were recorded, but 

also sand, algal turf etc. This does not seem obvious enough. How many observations per transect

were there? How were they pooled?

l104-105 2-tailed or 1-tailed t-test? Please clarify.

l105-106 Please explain the PERMANOVA design. What similarity measure was used? What R version, what package, what function, what factors in the model? Did you perform PERMDISP to check for homogeneity of variances? This is important, since significant PERMANOVA results can be caused by the difference in within-group dispersion. What variables were included in the analysis? Only live species covers, or also sand and non-coral substrate? Most important, you data (multiple transects with two measures per transect in 2000 and 2018) require the repeated measures design of the analysis with TransectID as a blocking random factor and Time as fixed effect with two levels. Any other analysis design looks inappropriate. This design also allows comparing community variation in space and time (see the comment to Figure 4 below).

l108-110 Matched Pair analysis in JMP pro is in fact the same paired Student t-test, which I found out from the manual for this software. In fact, t-test is inappropriate for percent covers, since they are never normally distributed (at least, the authors report no normality check or normalizing transformations of the data). Since you have proportional data with multiple zeros, you should consider either a simple non-parametric test, like Wilcoxon matched pairs test (less power), or check for zero-inflated Beta-distribution models for proportional data with zeros (more power).

l120-121 Reporting mixed sand and rubble along with live organisms is confusing and should be explained in Methods.

l124-125 It is not appropriate to provide only two temperature values for 2000 and 2018, you should report the whole long-term trend so that the reader could see the interannual variation and relate the observed 0.8 degree difference to other years. The temperature could rise slowly for 18 years or show interannual variation far exceeding 0.8 degrees, this should be clarified (consider adding a figure).

l125-126 PERMANOVA design unclear. Should be repeated measures two-factor with Transect as a random blocking one. See above.

l135-138 Here and below: averages should be given +-S.E. or S.D.

l137-138 Kappaphycus alvarezii cover changed from 0.33 to 0%, marginally significant on t-test, 

would be probably insignificant with Wilcoxon, probably no need to report as a change.

l162 Comparing how? Any data, any references? Comparison with other reefs looks purely speculative without the exact figures.

l162-164 Looks contradictory "We predict the reef will show the same resilience as other Hawaiian reefs"

l166-168 "Daily average summer temperature increased by 0.8° C in Kāneʻohe Bay from 2000 to 2018. This increase indicates corals at the Malauka`a fringing reef experienced increased temperature stress in 2018 than they did during the initial 2000 survey" -- comparing just two summers is inappropriate here, see above.

l170-171 This looks like Results, not Discussion.

l203-205 This (i.e. what exactly were the categories recorded along the transects) should be explained in Methods, maybe with photos like Figure 5 with arrows pointing at different categories. Authors should keep in mind that Methods must explain the procedure in such detail that would allow reproducing the survey.

l246-247 The possibility of misidentification makes the whole statement way too speculative.

l257-258 This belongs to Results, not Discussion. Also, more detail on sampling procedure needed to understand the odds of missing a rare species.

l263-270 I doubt that the result of cover change from 0 to 0.5% would remain significant and thus worth discussing when appropriate (i.e. non-parametric) test applied to covers.

l271-272 See above for my concerns about the way the temperature increase is reported.

Figure 2 NMDs figure should show no points for samples, not only ellipses (and/or centroids). Accurate stress value is needed instead of "<0.2". I would expect species scores on a separate side plot for clarity.

Figure 4 The legend not readable, a larger figure with larger pie-charts and larger legend needed, maybe 2 separate figures or plates. Please, specify how exactly the data was aggregated, i.e. how many transects per pie-chart were pooled. This figure is actually very important, allowing to relate the variation in assemblages in space and time, but almost nothing about this is in the Results and Discussion (also see my comments on the PERMANOVA design above).

Round 2

Reviewer 3 Report

The authors addressed some of my comments, but there are still major issues with the paper, namely unexpectedly ending Introduction and multivariate analysis. In its present form Introduction suddenly states no aim of the study, no hypothesis tested and somewhat ends unexpectedly after study site description have been transfered to Methods. Multivariate comparison of the communities is still inappropriate. The authors changed PERMANOVA to MANOVA, but do not show any appropriate design with Transect as a nested random effect, also they applied MANOVA to cover data without any normalizing transformations and checking any MANOVA assumptions like data normality and variance heterogeneity. Also, authors still do not provide an nMDS plot with samples displayed, only variation ellipses.

l112-113 "A paired two-tailed t-test was used to determine changes in summer temperatures between 113 years within RStudio [30]" - I do not understand what temperatures were actually compared with a paired test: all the summer long day-by-day? What was the sample size? This should be clarified. In fact, there is no clear trend in local annual temperature changes as seen on Figure 2. 

l114-115 and Figure 3. Like in my previous review, I insist on showing individual samples on nMDS plot, so that the variation would not be obscured, not only ellipses. Additional plot is probably needed for clarity.

l115-117 The authors changed PERMANOVA, which is appropriate for percent cover data to MANOVA, which is probably not, according to its assumptions (see below). In my previous review I pointed that PERMANOVA has inappropriate analysis design, not that the usage of PERMANOVA is wrong. The authors should utilize PERMANOVA with a blocking random factor Transect nested in fixed factor Year with two levels. The design of MANOVA they used is not explained enough, there is no information on nesting of the factors and fixed/random effects, and meeting the assumptions of MANOVA is not mentioned (normality, homogeneity of variances). Assumptions of MANOVA, however, can hardly be met with covers data with multiple zeros. Transforming the data (see angular transformation) could possibly help. PERMANOVA, however, even when properly designed, also needs a check for variance heterogeneity (PERMDISP). In its present way multivariate community comparison should be either dropped from the study (which may render it rather useless) or performed properly (consider PRIMER-E software package for PERMANOVA on mixed models or, if going forward with MANOVA, mixed linear models in R, but the data would need normalization and checked for homogeneity of variances). It does not look that the software package authors currently use is flexible enough to analyze the model with Transect as a nested random effect.

l134-135. This is not a proper way to report MANOVA or PERMANOVA results. You should clearly show the design, sums of squares by factor, F or pseudo-F, residuals, and degrees of freedom, not only p-value, so that the reader could see the design and contributions of different sources of variation into the model.
